# CONTINUAL MEMORY: CAN WE REASON AFTER LONG-TERM MEMORIZATION?

## ABSTRACT

Existing reasoning tasks often follow the setting of "end-to-end reasoning", which has an important assumption that the input contents can be always accessed while reasoning. However, human beings frequently adopt another reasoning setting in daily life, referred to "reasoning after memorizing". Concretely, human beings have the ability to unconsciously memorize their experiences within limited memory capacity, from which they can recall and respond to subsequent tasks. In this setting, the input contents are no longer available during reasoning, thus we need to compress and memorize the input stream in one pass, trying to answer general queries that are unseen before. Memory augmented neural networks introduce a write-read memory to perform such human-like memorization and reasoning, but they continually update the memory from current information and inevitably forget the early contents, failing to answer the queries relevant to early information. In this paper, we propose the Continual Memory (CM) to explore this ability of reasoning after long-term memorization. To alleviate the gradual forgetting of early information, we develop self-supervised memorization training with item-level and sequence-level objectives. We demonstrate several interesting characteristics of our continual memory via synthetic data, and evaluate its performance by several downstream tasks, including long-term text QA, long-term video QA and recommendation with long sequences.

## 1 INTRODUCTION

In recent years, the tremendous progress of neural networks has enabled machines to perform reasoning given a query $Q$ and the input contents $X$, e.g., infer the answer of given questions from the text/video stream in text/video question answering (Seo et al., 2016; Le et al., 2020b), or predict whether a user will click the given item based on the user behavior sequence in recommender systems (Ren et al., 2019; Pi et al., 2019). Studies that achieve top performances at such reasoning tasks usually follow the setting of "end-to-end reasoning", where the raw input contents $X$ is available at the time of answering $Q$. In this setting, complex interaction between $X$ and $Q$ can be designed to extract query-relevant information from $X$ with little loss, such as co-attention interaction (Xiong et al., 2016). Though these methods (Seo et al., 2016; Le et al., 2020b) can effectively handle these reasoning tasks, they require unlimited storage resources to hold the original input $X$. Further, they have to encode the whole input and develop the elaborate interaction from scratch, which are time consuming. This is not acceptable for online services that require instant response such as recommender systems, as the input sequence becomes extremely long (Ren et al., 2019).

Another setting of "reasoning after memorization", which has the restrictions that the raw input $X$ is *not* available at the time of answering $Q$, requires the model to first digest $X$ in a streaming manner, i.e., incrementally compress the current subsequence of $X$ into a memory $M$ with very limited capacity (size much smaller than $|X|$). Under such constraints, in the inference phase, we can only capture query-relevant clues from the limited states $M$ (rather than $X$) to infer the answer to $Q$, where the information compression procedure in $M$ is totally not aware of $Q$, posing great challenges of what to remember in $M$. This setting is very similar to the daily situation of our human beings, i.e., we may not even know the tasks $Q$ that we will answer in the future when we are experiencing current events, and we also cannot go back to replay when we are solving problems at hand. However, it's our instincts, which continually process information during our entire life with

limited and compressed memory storages, that allow us to recall and draw upon past events to frame our behaviors given the present situations (Moscovitch et al., 2016; Baddeley, 1992).

Compared to "end-to-end reasoning", "reasoning after memorization" though may not achieve better precisions at regular tasks with short sequences according to literatures (Park et al., 2020), is naturally a better choice for applications like long-sequence recommendation (Ren et al., 2019) and long-text understanding (Ding et al., 2020). Maintaining $M$ can be incremental with only a small part of inputs at each timestep while inference over $M$ and $Q$ is also tractable for online service. Memory augmented neural networks (MANNs) (Graves et al., 2014; 2016) introduce a write-read memory that already follows the setting of "reasoning after memorization", which compress the input contents into a fixed-size memory and only read relevant information from the memory during reasoning. However, existing works do not emphasize on using MANNs to perform long-term memory-based reasoning. They learn how to maintain the memory only by back-propagated losses to the final answer and do not design specific training target for long-term memorization, which inevitably lead to the gradual forgetting of early contents (Le et al., 2019a). That is, when dealing with the long-term input stream, the memory may only focus on current contents and naturally neglect long-term clues. Thus, existing MANNs fail to answer the query relevant to early information due to the lack of long-term memorization training.

In this paper, we propose the Continual Memory (CM) to further explore this ability of reasoning after long-term memorization. Specifically, we compress the long-term input stream into the continual memory with fixed-size capacity and infer subsequent queries based on the memory. To overcome gradual forgetting of early information and increase the generalization ability of the memorization technique, we develop the extra self-supervised task to recall the recorded history contents from the memory. This is inspired by the fact that human beings can recall details nearby some specific events and distinguish whether a series of events happened in the history, which respectively correspond to two different memory process revealed in cognitive, neuropsychological, and neuroimaging studies, i.e., *recollection* and *familiarity* (Yonelinas, 2002; Moscovitch et al., 2016). Concretely, we design the self-supervised memorization training with item-level and sequence-level objectives. The item-level objective aims to predict the masked items in history fragments, which are sampled from the original input stream and parts of items are masked as the prediction target. This task tries to endow the *recollection* ability that enables one to relive past episodes. And the sequence-level objective tries to distinguish whether a historical fragment ever appears in the input stream, where we directly sample positive fragments from the early input stream and replace parts of the items in positive ones as negative fragments. This task enables the *familiarity* process that can recognize experienced events or stimulus as familiar. We also give implementations on segment-level maintenance of memory to better capture context clues and improve the modeling efficiency. We illustrate the long-term memorization ability of our continual memory via a synthetic task, and evaluate its performance at solving real-world downstream tasks, including long-term text QA, long-term video QA and recommendation with long sequences, showing that it achieves significant advantages over existing MANNs in the "reasoning after memorizing" setting.

## 2 RELATED WORKS

Memory Augmented Neural Networks (MANNs) introduce external memory to store and access the past information by differentiable write-read operators. Neural Turing Machine (NTM) (Graves et al., 2014) and Differentiable Neural Computer (DNC) (Graves et al., 2016) are the typical MANNs for human-like reasoning under the setting of "reasoning after memorizing", whose inference relies only on the memory with limited capacity rather than starting from the original input data. In this line of research, Rae et al. (2016) adopt the sparse memory accessing to reduce computational cost. Csordás & Schmidhuber (2019) introduce the key/value separation problem of content-based addressing and adopt a mask for memory operations as a solution. Le et al. (2019b) manipulate both data and programs stored in memory to perform universal computations. And Santoro et al. (2018); Le et al. (2020a) consider the complex relational reasoning with the information they remember.

However, these works exploit MANNs mainly to help capture long-range dependencies in dealing with input sequences, but not paying efforts in dealing with the gradual forgetting issue in MANNs (Le et al., 2019a). They share the same training objective as those methods developed for the setting of "end-to-end reasoning", inevitably incurring gradual forgetting of early contents (Le

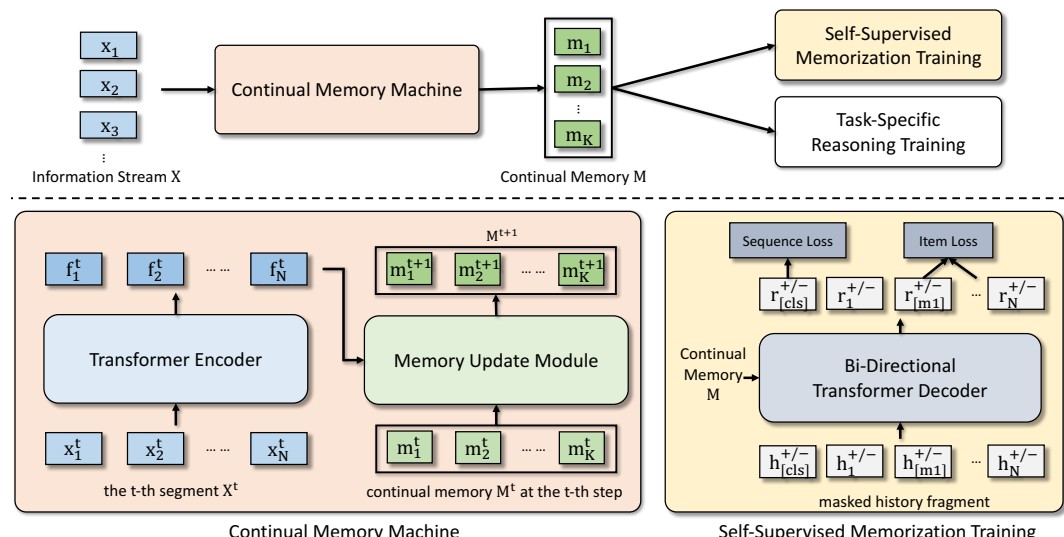

Figure 1: The Framework of Continual Memory and Self-Supervised Memorization Training.

et al., 2019a). Recently, there are a few works trying to alleviate this problem in training MANNs. Le et al. (2019a) propose to measure "remember" ability by the final gradient on the early input, and adopt a uniform writing operation on the memory. Rae et al. (2019) minimize the difference between successive memory states along with the reasoning objective, as they assume the steadily changed memory states will benefit remembering older information. And Munkhdalai et al. (2019) design the meta-learned neural memory instead of the conventional array-structured memory and memorize the current and past information by reconstructing the written values via the memory function. Note that our approach is different and parallel to these techniques, since we give no assumptions on what behavior will remember the most. Instead, we optimize towards this goal directly by designing auxiliary tasks in a self-supervised manner. A recent work (Park et al., 2020) also introduces a self-supervised memory loss to ensure how well the current input is written to the memory, but it only focuses on remembering the current information and ignoring the long-term information forgetting.

Continual learning (Kirkpatrick et al., 2017; Lopez-Paz & Ranzato, 2017; Chaudhry et al., 2018; de Masson d'Autume et al., 2019) is another field about the forgetting problem of neural networks, which aims to learn from an infinite stream of data and gradually extend acquired knowledge without catastrophic forgetting of early knowledge. But our continual memory focuses on remembering the infinite information stream and try to overcome gradually forgetting of long-term information.

# 3 CONTINUAL MEMORY

## 3.1 PROBLEM FORMULATION

Given the input stream $\mathbf{X} = \{\mathbf{x}_1, \mathbf{x}_2, \cdots\}$ and a query $\mathbf{Q}$, the methods under the setting of "end-to-end reasoning" directly learn the reason model $\mathcal{T}(\mathbf{X}, \mathbf{Q})$ to predict the answer $\mathbf{A}$. These is an important assumption that the input stream $\mathbf{X}$ can be always accessed while reasoning. And complex interaction between $\mathbf{X}$ and $\mathbf{Q}$ can be designed to extract query-relevant information in $\mathcal{T}(\mathbf{X}, \mathbf{Q})$. Obviously, these methods have to store the original input and infer the answer from scratch when the query is known. But under the setting of "reasoning after memorizing", we compress the input stream $\mathbf{X}$ into a fixed-size memory $\mathbf{M} = \{\mathbf{m}_k\}_{k=1}^{K}$ with $K$ memory slots and then infer the answer for any relevant query $\mathbf{Q}$ by $\mathbf{A} = \mathcal{R}(\mathbf{M}, \mathbf{Q})$. Here we only need to store the compressed memory $\mathbf{M}$, which can be updated in real-time and reused for a series of queries. Since the slot number $K$ in the memory is irrelevant to the input length $|X|$, this setting only requires $O(1)$ storage space rather than $O(|X|)$ in the setting of "end-to-end reasoning".

## 3.2 CONTINUAL MEMORY MACHINE

As shown in Figure 1, given the input stream $\mathbf{X} = \{\mathbf{x}_1, \mathbf{x}_2, \cdots\}$, we apply a continual memory machine $\mathcal{G}_\Theta(\cdot)$ to compress them into continual memory $\mathbf{M} = \{\mathbf{m}_k\}_{k=1}^K$ with $K$ memory slots. By self-supervised memorization training, we try to overcome the gradual forgetting of early information and make it possible to capture the clues at any time in the stream. Concretely, based on continual memory, we develop the history recall model $\mathcal{H}_\xi(\cdot)$ to reconstruct the masked history fragments and distinguish positive history fragments from negative ones. Simultaneously, we train the task-specific reason model $\mathcal{R}_\Omega(\cdot)$ based on continual memory. Under the setting of "reasoning after memorizing", we can develop the continual memory $\mathbf{M} = \mathcal{G}_\Theta(\mathbf{X})$ and then infer the answer for any relevant query $\mathbf{Q}$ by $\mathbf{A} = \mathcal{R}_\Omega(\mathbf{M}, \mathbf{Q})$.

We deal with the input stream $\mathbf{X}$ from the segment level rather than item level, i.e., we cut the input sequence into fixed-length segments and memorize them into the memory slots segment-by-segment. Compared to existing MANNs (Graves et al., 2014; 2016), which store the input stream item-by-item orderly with a RNN-based controller, our segment-level memorization can further capture the bidirectional context of each item and improve the modeling efficiency. We denote the $t$-th segment as $\mathbf{X}^t = \{\mathbf{x}_n^t\}_{n=1}^N$ and the current memory as $\mathbf{M}^t = \{\mathbf{m}_k^t\}_{k=1}^K$, where we have recorded $t$-1 segments in $\mathbf{M}^t$. The $\mathbf{x}_n^t$ and $\mathbf{m}_n^t$ have the dimension $d_x$ and $d_m$, respectively.

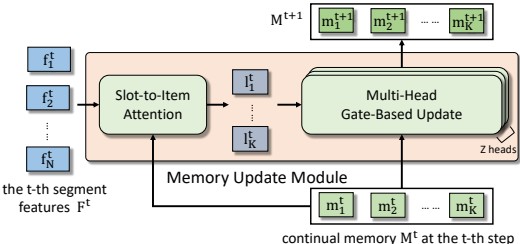

Figure 2: The Process of Memory Updating.

We first model the $t$-th segment by a Transformer encoder (Vaswani et al., 2017) and obtain the sequence features $\mathbf{F}^t = \{\mathbf{f}_n^t\}_{n=1}^N$ with dimension $d_x$. After it, we apply a memory update module to write $\mathbf{F}^t$ into $\mathbf{M}^t$. As shown in Figure 2, we apply a slot-to-item attention to align the sequence features to slot features in the current memory $\mathbf{M}^t$, and then develop multi-head gate-based update. Concretely, we first calculate the slot-to-item attention matrix where each element means the relevance of a slot-item pair, and then learn aligned features $\mathbf{L}^t = \{\mathbf{l}_k^t\}_{k=1}^K$ for each slot, given by

$$\alpha_{kn}^t = \mathbf{w}_a^\top \tanh(\mathbf{W}_1^a \mathbf{m}_k^t + \mathbf{W}_2^a \mathbf{f}_n^t + \mathbf{b}^a), \ \hat{\alpha}_{kn}^t = \frac{\exp(\alpha_{kn}^t)}{\sum_{j=1}^K \exp(\alpha_{jn}^t)}, \ \mathbf{l}_k^t = \sum_{n=1}^N \hat{\alpha}_{kn}^t \mathbf{f}_n^t, \qquad (1)$$

where $\mathbf{W}_1^a \in \mathbb{R}^{d_{model} \times d_m}$, $\mathbf{W}_2^a \in \mathbb{R}^{d_{model} \times d_x}$ and $\mathbf{b}^a \in \mathbb{R}^{d_{model}}$ are the projection matrices and bias. $\mathbf{w}_a^\top$ is the row vector. After it, we project slot features and aligned features into $Z$ subspaces, which is similar to the Multi-Head setting in Transformer (Vaswani et al., 2017), given by

$$\mathbf{m}_{kz}^t = \mathbf{W}_z^M \mathbf{m}_k^t, \ \mathbf{l}_{kz}^t = \mathbf{W}_z^L \mathbf{l}_k^t, \qquad (2)$$

where $\mathbf{W}_z^M \in \mathbb{R}^{d_{model}/Z \times d_m}$ and $\mathbf{W}_z^L \in \mathbb{R}^{d_{model}/Z \times d_x}$ are the projection matrices. The $\mathbf{m}_{kz}^t$ and $\mathbf{l}_{kz}^t$ are the slot and aligned sub-features in the $z$-th subspace. Next, the $k$-th slot sub-feature $\mathbf{m}_{kz}^t$ is updated with the corresponding sub-feature $\mathbf{l}_{kz}^t$ based on the $z$-th GRU unit with $\frac{d_{model}}{Z}$-d hidden states, given by

$$\mathbf{m}_{kz}^{t+1} = \mathrm{GRU}_z(\mathbf{m}_{kz}^t, \mathbf{l}_{kz}^t), \qquad (3)$$

where $\mathbf{l}_{kz}^t$ is the current input of the $z$-th GRU unit. Next, the new slot feature $\mathbf{m}_k^{t+1}$ is aggregated from $Z$ subspaces by $\mathbf{m}_k^{t+1} = \mathbf{W}^o \mathrm{Concat}(\mathbf{m}_{k1}^{t+1}, \cdots, \mathbf{m}_{kZ}^{t+1})$, where $\mathbf{W}^o \in \mathbb{R}^{d_m \times d_{model}}$ is the aggregation matrix. After the memorization of $T$ segments, we can obtain continual memory $\mathbf{M}^T$ and we denote it by $\mathbf{M}$ for convenience. Note that, at the inference stage, we can develop and update the continual memory in real time by $\mathcal{G}_\Theta(\cdot)$, thus we do not need the input contents during subsequent reasoning and have the ability to reason after long-term memorization.

## 3.3 SELF-SUPERVISED MEMORIZATION TRAINING

After memorizing the input stream, we conduct self-supervised memorization training to alleviate the gradual forgetting of early information by the history recall model $\mathcal{H}_\xi(\cdot)$ with item-level and sequence-level objectives, where the item-level objective aims to reconstruct the masked positive

history fragments and the sequence-level objective tries to distinguish positive history fragments from negative ones.

Concretely, we sample the preceding segment from the input stream as the positive history fragment $\{h_1^+, h_2^+, \cdots, h_N^+\}$ with $N$ items, where each item $h_*^+$ corresponds to a feature $\mathbf{x}_*$ in the stream. We then mask 50% of items in the fragment and add an especial item [cls] at the beginning to obtain the masked positive history fragment $H^+ = \{h_{[cls]}^+, h_1^+, h_{[mask_1]}^+, \cdots, h_N^+\}$. In order to guarantee that the model $\mathcal{H}_\xi(\cdot)$ reconstructs the masked fragment by utilizing continual memory rather than only relying on fragment context, we set the mask ratio to 50% instead of 15% in BERT (Devlin et al., 2019). Moreover, we construct the masked negative history fragment $H^- = \{h_{[cls]}^-, h_1^-, h_{[mask_1]}^-, \cdots, h_N^-\}$ by replacing 50% of unmasked items in the positive fragments, where the replacement items are sampled from other input stream to make the negative fragments distinguishable. Here we construct $B$ masked positive fragments with corresponding $B$ negative ones. Next, we adopt a bidirectional Transformer decoder (Vaswani et al., 2017) without the future masking to model each history fragment $H^+/H^-$. In the decoder, each history item can interact with all other items. The continual memory $\mathbf{M}$ is input to the "encoder-decoder multi-head attention sub-layer" in each decoder layer, where the queries come from the previous decoder layer and the memory $\mathbf{M}$ are regarded as the keys and values. This allows each item in the decoder to attend over all slot features in the memory. Finally, we obtain the features $\{\mathbf{r}_{[cls]}^{+/-}, \mathbf{r}_1^{+/-}, \mathbf{r}_{[mask_1]}^{+/-}, \cdots, \mathbf{r}_N^{+/-}\}$ where each $\mathbf{r}_*^{+/-}$ has the dimension $d_x$.

**Item-Level Reconstruction.** We first predict the masked items of positive history fragments to build the item-level loss. Considering there are too many item types, we apply the contrastive training He et al. (2020); Chen et al. (2020) based on the ground truth and other sampled items. For the $N/2$ masked items, we compute the item-level loss by

$$\mathcal{L}_{item} = -\frac{2}{N} \sum_{i=1}^{N/2} \log \frac{\exp(\phi(\mathbf{r}_{[mask_i]}^+, \mathbf{y}_i))}{\exp(\phi(\mathbf{r}_{[mask_i]}^+, \mathbf{y}_i)) + \sum_{j=1}^{J} \exp(\phi(\mathbf{r}_{[mask_i]}^+, \mathbf{y}_j))}, \tag{4}$$

where $\mathbf{y}_i \in \mathbb{R}^{d_x}$ is the feature of ground truth of the $i$-th masked item, $\mathbf{y}_j \in \mathbb{R}^{d_x}$ is the feature of sampled items and $\phi(\cdot)$ is the inner product.

**Sequence-Level Prediction.** Next, we predict whether the masked history fragment ever appears in the current input stream, i.e. distinguish positive history segment from negative ones. Concretely, we project the feature $\mathbf{r}_{[cls]}^{+/-}$ into a confident score $s^{+/-}$ and develop the sequence-level loss by

$$\mathcal{L}_{seq} = -\sum_{i=1}^{B} \log(s_i^+) + \sum_{j=1}^{B} \log(1 - s_j^-), \tag{5}$$

where $B$ is the number of positive and negative history fragments.

### 3.4 TASK-SPECIFIC REASONING TRAINING

Besides self-supervised memorization training, we simultaneously develop task-specific reasoning training. For several downstream tasks, we propose different task-specific reason model $\mathcal{R}_\Omega(\mathbf{M}, \mathbf{Q})$ for any query $\mathbf{Q}$ based on continual memory $\mathbf{M}$. Here we adopt the simple and mature components in the reason model for fair comparison. The details are introduced in Appendix A.1. Briefly, we first learn the query representation $\mathbf{q}$ by a task-specific encoder and then perform the multi-hop attention-based reasoning. Finally, we obtain the reason loss $\mathcal{L}_r$ from $\mathcal{R}_\Omega(\mathbf{M}, \mathbf{Q})$.

Eventually, we combine the memorization and reason losses to train our model, given by

$$\mathcal{L}_{cm} = \lambda_1 \mathcal{L}_{item} + \lambda_2 \mathcal{L}_{seq} + \lambda_3 \mathcal{L}_r, \tag{6}$$

where $\lambda_1$, $\lambda_2$ and $\lambda_3$ are applied to adjust the balance of three losses.

## 4 EXPERIMENTS

We validate our continual memory on synthetic data and several downstream tasks, including long-term text question answering, long-term video question answering and recommendation with long sequences.

### 4.1 EXPERIMENT SETTING

**Model Setting**. We first introduce the common model settings for all downstream tasks. We set the layer number of the Transformer encoder and bi-directional Transformer decoder to 3. The head number in Multi-Head Attention is set to 4. And the subspace number $Z$ in the memory update module is also set to 4. We set $\lambda_1$, $\lambda_2$ and $\lambda_3$ to 1.0, 0.5 and 1.0, respectively. The number $B$ of history fragments is set to 5. During training, we apply an Adam optimizer (Duchi et al., 2011) to minimize the multi-task loss $\mathcal{L}_{cm}$, where the initial learning rate is set to 0.001.

**Baseline.** We compare our continual memory with the end-to-end reasoning methods and the memory-based reasoning approaches under the "reasoning after memorizing" setting. The end-to-end baselines are different in downstream tasks and the memory-based baselines mainly are DNC (Graves et al., 2016), NUTM (Le et al., 2019b), STM (Le et al., 2020a) and DMSDNC (Park et al., 2020). For fair comparison, we modify the reason module of memory-based baselines to be consistent with our continual memory, i.e. we conduct multi-hop attention-based reasoning based on the built memory. And the number of memory slots in these baselines is also set to K. Besides, we set the core number of NUTM to 4, the query number of STM to 8 and the memory block number of DMSDNC to 2.

### 4.2 SYNTHETIC EXPERIMENT

**Synthetic Dataset.** We first introduce the setting of the synthetic task. Here we abstract the general concepts of reasoning tasks (QA/VQA/Recommendation) to construct the synthetic task. We define the input sequence as a **Stream** and each item in the sequence as a **Fact**, where the stream and fact can correspond to the text sequence and word token in text QA. We set the number of fact types to $R_f$, that is, each fact can be denoted by a $R_f$-d one-hot vector and obtain the fact feature by a trainable embedding layer. Different facts can correspond to different words in text QA. Considering reasoning tasks often need to retrieve vital clues related to the query from the given input and then infer the answer, we define the query-relevant facts in the stream as the **Evidence** and regard the **Evidence-Query-Answer** triple as the **Logic Chain**. As shown in Figure 3, given a stream and a query, we need to infer the answer if the stream contains the evidence. Specifically, we set the number of query types to $R_q$ and each query can be denoted by a $R_q$-d one-hot vector. For each query, we set the number of answer types to $R_a$. That is, there are totally $R_q * R_a$ query-answer pairs and we need to synthesize $R_q * R_a$ corresponding evidences of each pair. Concretely, each evidence is denoted by a sequence of facts $\{\text{fact}_1, \cdots, \text{fact}_{R_c}\}$, which orderly appear in the input stream. And $R_c$ is the length of the evidence. During the evidence synthesis, we first define 20 different group and uniformly split these facts and queries to 20 groups. Next, if a query belongs to group $k$, we randomly sample $R_c$ facts from the group as the evidence, and then assign the evidence to a query-answer pair to generate a fixed logic chain.

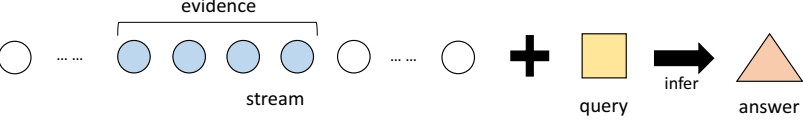

Figure 3: A Data Sample with Consecutive Evidence .

Eventually, we synthetic 400 data samples for each logic chain to train the models. Each sample contains the input stream with $R_l$ items, a query and an answer. Concretely, we first sample $R_l$ facts as a sequence and then place the evidence in the sequence, where we guarantee each stream-query pair corresponds to a unique answer. Moreover, we synthesize two datasets with the consecutive and discrete evidence, respectively. The facts in the consecutive evidence continuously appear in the stream, but facts in the discrete evidence are distributed in different parts of the stream, where we still make these facts exist in a certain interval that does not exceed 20% length of the input stream.

**Baselines and Model Details.** The **Directly Reason** method first models the input stream by RNN to obtain the stream feature, then concatenates the stream feature with the query feature and predicts the answer by a linear layer. The **Multi-Hop Reason** method further applies multiple attention layers after RNN-based stream modeling to capture the query-relevant clues. In the main experiment, we

Table 1: Performance Comparisons on Synthetic Data. $R_f$=400, $R_l$=200, $R_q$=40, $R_a$=30, $R_c$=5.

| Method | Setting | Discrete Evidence | | | Consecutive Evidence | | |
|---|---|---|---|---|---|---|---|
| | | Early | Later | All | Early | Later | All |
| Directly Reason | End-to-End | 9.45 | 9.32 | 9.39 | 13.57 | 13.41 | 13.49 |
| Multi-Hop Reason | End-to-End | 32.16 | 33.42 | 32.29 | 34.38 | 34.50 | 34.44 |
| DNC | Memory-Based | 13.37 | 22.54 | 17.95 | 20.56 | 26.59 | 23.58 |
| NUTM | Memory-Based | 17.84 | 23.59 | 20.72 | 24.31 | 29.71 | 27.01 |
| STM | Memory-Based | 17.90 | 23.47 | 20.68 | 23.55 | 29.64 | 26.59 |
| DMSDNC | Memory-Based | 18.17 | 24.21 | 21.19 | 24.92 | 30.74 | 27.83 |
| CM (Only Reason) | Memory-Based | 18.71 | **25.13** | 21.92 | 25.79 | 31.38 | 28.63 |
| CM (Full) | Self-Sup. Memory | **22.14** | 24.98 | **23.56** | **28.42** | **31.71** | **30.07** |
| CM (Only $\mathcal{L}_{item}$) | Self-Sup. Memory | 21.79 | 24.46 | 23.13 | 27.88 | 31.12 | 29.50 |
| CM (Only $\mathcal{L}_{seq}$) | Self-Sup. Memory | 19.75 | 23.81 | 21.78 | 26.63 | 30.25 | 28.44 |
| CM (Single Head) | Self-Sup. Memory | 21.24 | 24.12 | 22.68 | 27.79 | 31.14 | 29.47 |

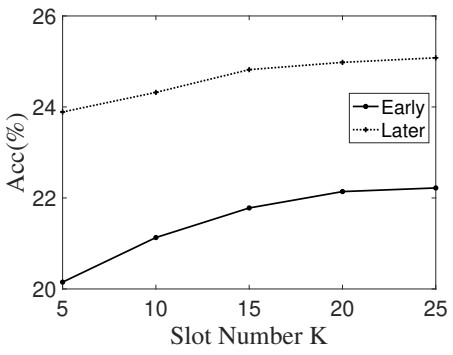

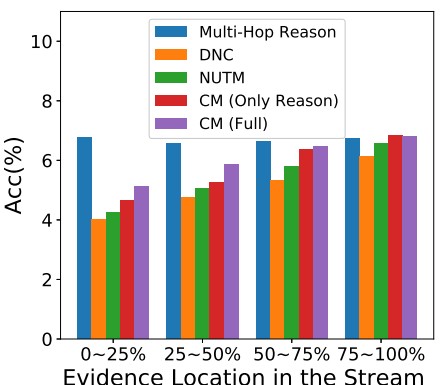

Figure 4: Effect of the Memory Slot Number.  Figure 5: Performance on Synthetic Data with Longer Stream and Evidence.

set the dataset hyper-parameters $R_f$, $R_l$, $R_q$, $R_a$ and $R_c$ to 400, 200, 40, 30, and 5, respectively. The facts of the evidence may appear in different stages of the input stream. **Early** means the facts appear in the preceding 50% of the stream and **Later** means the facts appear in the subsequent 50%. For our continual memory, we set the $d_x$, $d_m$ and $d_{model}$ to 128. The number K of memory slots and length N of segments are set to 20 and 10, respectively. And we sample all other facts as negative items in $\mathcal{L}_{item}$.

**Evaluation Results.** Table 1 reports the performance comparison between our method and baselines, where CM (Full) is the full model with the memorization and reasoning training and CM (Only Reason) only employ the task-specific reasoning training. Overall, end-to-end methods have the close early and later performance, but memory-based approaches DNC, NUTM, STM, DMS-DNC and CM (Only Reason) achieve the terrible early performance due to the gradual forgetting. By the self-supervised memorization training, our CM (Full) significantly improves the early accuracy and achieves the best memory-based reasoning performance. This fact suggests our proposed memorization training can alleviate the gradual forgetting of early information. Besides, CM (Only Reason) outperforms other memory-based methods, which indicates our continual memory machine can better memorize the long-term information even without the memorization training. Moreover, we can find the Directly Reason approach achieves the worst performance but the Multi-Hop Reason method has a high accuracy, which demonstrates the performance of end-to-end methods mainly depends on the complicated interaction between the input contents and queries.

**Ablation Study.** We next perform ablation studies on the memorization loss and multi-head update strategy. Concretely, we first remove the sequence-level or item-level loss to produce two ablation

models CM (Only $\mathcal{L}_{item}$) and CM (Only $\mathcal{L}_{seq}$). As shown in Table 1, CM (Full) outperforms two ablation models on all metrics, which illustrates two self-supervised losses are both helpful for long-term memorization. Moreover, CM (Only $\mathcal{L}_{item}$) achieves better accuracy than CM (Only $\mathcal{L}_{seq}$), demonstrating the importance of the item-level loss. Next, we discard the multi-head setting from the memory update module to generate the ablation model CM(Single Head). From the results, we can find the CM(Single Head) has the performance degradation, indicating the multi-head update can improve the memorization ability of continual memory.

**Hyper-Parameters Analysis.** We then explore the effect of the slot number $K$. We set $K$ to [5, 10, 15, 20, 25] and display the results in Figure 4. We note that the model performance gradually improves with the increase of slot number and it reaches the bottleneck when the slot number is set to 25. Besides, the early performance has more significant improvement than the later performance.

**Longer Stream and Evidence.** To further validate the characteristics of our continual memory, we synthesize a more complicated dataset where $R_f$, $R_l$, $R_q$, $R_a$ and $R_c$ are set to 2000, 1000, 40, 30, and 10, respectively. That is, the dataset contains the longer input stream and more complex evidence. We set the number K of memory slots and length N of segments to 20. As shown in Figure 5, we display the reasoning performance of each model when the facts in the evidence appear in different locations of the input stream. For example, 25∼50% means the facts appear between 25% and 50% of the stream. We can observe that our CM (Full) has an obvious performance improvement compared to memory-based methods in the 0∼25% and 25∼50% stage, but achieves a slight improvement in the 75∼100% stage. This verifies our self-supervised training is beneficial for long-term memorization.

Table 2: Performance Comparisons for Long-Term Text Question Answering on NarrativeQA.

| Method | Setting | Val | Test |
|---|---|---|---|
| AS Reader | End-to-End | 26.9 | 25.9 |
| E2E-MN | End-to-End | 29.1 | 28.6 |
| DNC | Memory-Based | 25.8 | 25.2 |
| NUTM | Memory-Based | 27.7 | 27.2 |
| STM | Memory-Based | 27.2 | 26.7 |
| DMSDNC | Memory-Based | 28.1 | 27.5 |
| CM (Only Reason) | Memory-Based | 28.0 | 27.6 |
| CM (Full) | Self-Sup. Memory | **28.8** | **28.1** |

Table 3: Performance Comparisons for Long-Term Text Question Answering on TVQA.

| Method | Setting | Acc. |
|---|---|---|
| Multi-Stream | End-to-End | 63.14 |
| MSAN | End-to-End | 65.77 |
| DNC | Memory-Based | 55.92 |
| NUTM | Memory-Based | 58.41 |
| CM (Only Reason) | Memory-Based | 59.87 |
| CM (Full) | Self-Sup. Memory | **61.12** |

## 4.3 LONG-TERM TEXT QUESTION ANSWERING

**Dataset, Baseline and Model Details.** We apply two multi-choice datasets NarrativeQA (Kočiský et al., 2018) and TVQA (Lei et al., 2018) for long-term text question answering. The two datasets provide the answer candidates thus they are suitable for the setting "reasoning after memorizing". Note that the TVQA dataset also provides the video contents as input but we only use the subtitles in the videos. For NarrativeQA, the **AS Reader** (Kadlec et al., 2016) applies a pointer network to generate the answer and **E2E-MN** (Sukhbaatar et al., 2015) employs the end-to-end memory network to conduct multi-hop reasoning. For TVQA, the **Multi-Stream** (Lei et al., 2018) method develops the query-subtitle interaction for reasoning. And **MSAN** (Kim et al., 2020) first localizes the clues relevant to the question and then predicts the answer. For our continual memory, we set the $d_x$, $d_m$ and $d_{model}$ to 256. The number K of memory slots and length N of segments are both set to 20. And we sample all other words as negative items in $\mathcal{L}_{item}$.

**Evaluation Results.** As shown in Table 2 and Table 3, the results are similar to synthetic experiments, i.e. our continual memory obtains the best performance among the memory-based approaches. And CM (Full) achieves the further improvement compared to CM (Only Reason), demonstrating the self-supervised memorization training can boost the reasoning ability of continual memory. Moreover, our proposed method achieves the accuracy close to the excellent end-to-end method E2E-MN in the NarrativeQA dataset, but still has a large performance gap with MSAN in the TVQA dataset. This is mainly because the MSAN method applies the two-stage reasoning with elaborate interactions between texts and queries.

### 4.4 LONG-TERM VIDEO QUESTION ANSWERING

**Dataset, Baseline and Model Details**: The ActivityNet-QA dataset (Yu et al., 2019) contains 5,800 videos from the ActivityNet (Caba Heilbron et al., 2015). The average video duration of this dataset is about 180s and is the longest in VQA datasets. We compare our method with three basic end-to-end baselines **E-VQA**, **E-MN**, **E-SA** from (Yu et al., 2019) and the SOTA end-to-end model **HCRN** (Le et al., 2020b). For our continual memory, we set the $d_x$, $d_m$ and $d_{model}$ to 256. The number K of memory slots and length N of segments are both set to 20. And in $\mathcal{L}_{item}$, we select 30 other frame features from the video as the sampled items.

**Evaluation Results.** As shown in Table 4, the CM (Only Reason) obtains the better performance than other memory-based models DNC and NUTM, and CM (Full) further achieves the 1.1% absolute improvement, showing the effectiveness of our model designs and self-supervised memorization training. Moreover, our method outperforms the basic end-to-end baselines and slightly worse than the SOTA method HCRN. This suggests our continual memory can reduce the gap between memory-based and end-to-end reasoning paradigms.

Table 4: Performance Comparisons for Long-Term Video Question Answering on ActivityNet-QA.

| Method | Setting | Acc. |
|---|---|---|
| E-VQA | End-to-End | 25.2 |
| E-MN | End-to-End | 27.9 |
| E-SA | End-to-End | 31.8 |
| HCRN | End-to-End | 37.6 |
| DNC | Memory-Based | 30.3 |
| NUTM | Memory-Based | 33.1 |
| CM (Only Reason) | Memory-Based | 34.6 |
| CM (Full) | Self-Sup. Memory | **35.7** |

Table 5: Performance Comparisons for Lifelong Sequence Recommendation on XLong.

| Method | Setting | AUC |
|---|---|---|
| GRU4REC | End-to-End | 0.8702 |
| Caser | End-to-End | 0.8390 |
| RUM | End-to-End | 0.8649 |
| DIEN | End-to-End | 0.8793 |
| HPMN | Memory-Based | 0.8645 |
| MIMN | Memory-Based | 0.8731 |
| CM (Only Reason) | Memory-Based | 0.8756 |
| CM (Full) | Self-Sup. Memory | **0.8824** |

### 4.5 LIFELONG SEQUENCE RECOMMENDATION

**Dataset, Baseline and Model Details.** The lifelong sequence recommendation aims to predict whether the user will click a given item based on long sequences, thus it can be regarded as a long-term reasoning task. The XLong dataset (Ren et al., 2019) is sampled from the click logs on Alibaba. The length of historical behavior sequences in this dataset is 1000. We compare our method with four end-to-end methods **GRU4REC** (Hidasi et al., 2015), **Caser** (Tang & Wang, 2018), **DIEN** (Zhou et al., 2019), **RUM** (Chen et al., 2018) and two memory-based methods **HPMN** (Ren et al., 2019) and **MIMN** (Pi et al., 2019), where the HPMN method builds the memory by hierarchical RNNs and the MIMN method introduces a write-read memory as in (Graves et al., 2014). For our continual memory, we set the $d_x$, $d_m$ and $d_{model}$ to 64. The number K of memory slots and length N of segments are both set to 20. And in $\mathcal{L}_{item}$, we select 200 items from the large item set as the sampled items.

**Evaluation Results.** As shown in Table 5, our CM (Full) method not only outperforms other memory-based approaches, but also achieves better performance than state-of-the-art end-to-end baselines. This is because our continual memory can aggregate and organize the long-term interests from user behavior sequences and these interests can be activated during next-item prediction. But the end-to-end approaches may fail to learn such informative interest representations.

## 5 CONCLUSIONS

In this paper, we propose the continual memory to explore the ability of reasoning after long-term memorization. We compress the input stream into continual memory with the self-supervised memorization training and task-specific reasoning training. Based on continual memory, we can capture the clues for the subsequent queries and give the correct responses. Extensive experiments on a series of downstream tasks verify the effectiveness of the continual memory. For future work, we will further explore the property of continual memory.

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

# A APPENDIX

## A.1 TASK-SPECIFIC REASON MODELS

In this section, we introduce the task-specific reason model $\mathcal{R}_\Omega(\mathbf{M}, \mathbf{Q})$, where $\mathbf{M}$ is the built memory and $\mathbf{Q}$ is the given query. Specifically, we first model the query feature $\mathbf{q} \in \mathbb{R}^{d_{model}}$ by a task-specific encoder. For synthetic experiments, the given query $\mathbf{Q}$ is a one-hot vector and we directly obtain $\mathbf{q}$ by an embedding layer. For long-term text and video QA tasks, the query $\mathbf{Q}$ is a sentence and we apply a bi-directional GRU to learn the sentence feature $\mathbf{q}$. As for the recommendation task with long sequences, the given query is a target item with the unique id and we likewise learn an embedding layer to obtain the feature $\mathbf{q}$.

Next, we develop the multi-hop attention-based reasoning on continual memory $\mathbf{M}$. Concretely, at each step $c$, we capture the importance memory feature $\mathbf{e}^c \in \mathbb{R}^{d_m}$ from $\mathbf{M}$ based on the current query $\mathbf{q}^{c-1}$ using an attention method, given by

$$\gamma_k^c = \mathbf{w}_c^\top \tanh(\mathbf{W}_1^c \mathbf{q}^{c-1} + \mathbf{W}_2^c \mathbf{m}_k + \mathbf{b}^c), \ \hat{\gamma}_k^c = \frac{\exp(\gamma_k^c)}{\sum_{j=1}^K \exp(\gamma_j^c)}, \ \mathbf{e}^c = \sum_{k=1}^K \hat{\gamma}_k^c \mathbf{m}_k,$$

where $\mathbf{W}_1^c \in \mathbb{R}^{d_{model} \times d_{model}}$, $\mathbf{W}_2^c \in \mathbb{R}^{d_{model} \times d_m}$ and $\mathbf{b}^c \in \mathbb{R}^{d_{model}}$ are the projection matrices and bias. And $\mathbf{w}_c^\top$ is the row vector. We then produce the next query $\mathbf{q}^c = \mathbf{W}^q[\mathbf{e}^c; \mathbf{q}^{c-1}] \in \mathbb{R}^{d_{model}}$, where $\mathbf{W}^q \in \mathbb{R}^{d_{model} \times (d_m + d_{model})}$ is the projection matrix and $\mathbf{q}^0$ is the original $\mathbf{q}$. After C steps, we obtain the reason feature $\mathbf{q}^C$. The hyper-parameter $C$ is set to 2, 2, 2 and 1 for synthetic experiments, text QA, video QA and sequence recommendation, respectively.

After it, we design the final reasoning layer for different tasks. For synthetic experiments and long-term video QA with fixed answer sets, we directly apply a classification layer to select the answer and develop the cross-entropy loss $\mathcal{L}_r$. But text QA datasets NarrativeQA and TVQA provide different candidate answers for each query, we first model each candidate feature $\mathbf{a}_i$ by another bi-directional GRU and then concatenate $\mathbf{a}_i$ with $\mathbf{q}^C$ to predict the conference score for each candidate. Finally, we also learn the cross-entropy loss $\mathcal{L}_r$ based on answer probabilities. As for the sequence recommendation task, we can directly compute a confidence score based on $\mathbf{q}^C$ by a linear layer and build the binary loss function $\mathcal{L}_r$.

## A.2 HYPER-PARAMETER ANALYSIS OF SEGMENT LENGTH

We explore the effect of the segment length $N$ in the main experiment of the synthesis task. We set $N$ to [5, 10, 15, 20] and display the results in Figure 6. We can find that when the segment length is set to 5, the model achieves a terrible result and the performance is relatively stable when the length changes between 10 and 20. This is because when the segment is too short, important evidence may be scattered in different segments, and the model cannot effectively capture the evidence and infer the answer.

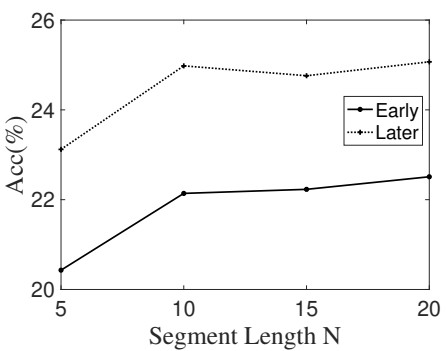

Figure 6: Effect of the Segment Length.

